# Sustainability Transitions in the Construction Sector: A Bibliometric Review

Luis Felipe Cândido [ID], Jose Carlos Lazaro *[ID], Adriano Olivier de Freitas e Silva [ID] and José de Paula Barros Neto *

Postgraduation Program on Managament and Controlling—PPAC, Faculty of Economics, Administration and Accounting, Campus Benfica, Federal University of Ceará—UFC, Av. Da Universidade 2431, Benfica, Fortaleza 60181-020, Brazil; luisfcandido2015@gmail.com (L.F.C.); adrianofrs@frn.uespi.br (A.O.d.F.e.S.)
* Correspondence: lazaro.ufc@gmail.com (J.C.L.); jpbarros@ufc.br (J.d.P.B.N.)

**Abstract:** Sustainability transition constitutes an important topic in innovation studies that have been providing insights into contemporary sustainability issues. These insights can help us to rethink how the construction industry can become more sustainable. Thus, this study review comprehensively analyzes the scientific production of ST in the CI through bibliometric analysis, using a sample of 121 documents from the Web of Science and Scopus databases. The review identified the evolution of scientific production and the top journals, institutions, nations, and authors contributing to this field and highlights a significant increase in publications since 2017. The VOSviewer was used to perform the science mapping and revealed the ongoing fragmentation within the publication network in the field. The bibliographic coupling and author keyword co-occurrence networks shed light on the research trends and directions. In sum, the scientific production on the transition to sustainability in the construction sector is diverse but relatively recent, indicating that the field is still in its early stages and requires more research for a comprehensive understanding of the subject. Overall, this study contributes by providing insightful information about the current state of TS in the CI, enabling dialogue between academic communities and stimulating interest in TS among those who have not yet addressed these issues.

**Keywords:** bibliometric analysis; construction projects; innovation; sociotechnical transition

## 1. Introduction

The Construction Industry (CI) significantly contributes to the aggregate economic activity in both developed and emerging economies [1]. Despite its relevance, the sector is globally recognized for its conservative attitude to the adoption of innovative sustainable technologies [2], its operational methods that are labor intensive at the construction site [3], and its low-tech intensity [4]. These main characteristics are manifested in high rates of waste [5,6], high rates of consumption of raw materials, and environmental pollution [6,7]. It is estimated that approximately 30% of global energy consumption and $CO_2$ emissions originate from the construction sector [8].

Against this background, the sector can be seen as strategic to sustainability transition due to its extensive interrelation with societal activities [9,10]. The built-up environment is where societal life materializes, thus its design and materials can influence the actions and practices that are inherent to an individual's daily life and the functioning of society [11]. This view enables us to perceive a building as the fundamental unit generating sustainability. Moreover, the construction sector represents one of the three key sectors to address the challenges of climate change for the European Union [12].

Thus, awareness of the importance of more sustainable construction projects has been increasing [13], leading to a rising amount of research in various fields of study, including those focused on innovation and technology for sustainability, particularly in the field of sustainability transitions. The transition to sustainability is one of the most prominent themes within studies on sociotechnical transitions [14] and consists of a set of approaches

to understand and support moving society towards sustainability [15]. One of the main focuses is to investigate how innovations can be incorporated or even become dominant in a given context, sometimes systemically modifying the current sociotechnical system and other times merely reconfiguring it [16].

Despite its acknowledged importance, ST studies in the CI remain insufficiently explored. The literature is recent, multidisciplinary, fragmented, and not widely spread within the construction research community. The research on sustainability in construction generally remains focused at the construction project level, which focuses on the field of engineering and adopts a predominantly technical approach [17]. As a result, a mismatch is present between sustainability studies and advancements in the sector toward a more sustainable production and consumption standard. This mismatch demands studies that not only explore the social, economic, and environmental impacts of the sector but also address institutional dynamics and broader social aspects as their subjects, such as those investigating sociotechnical transitions to sustainability, which support the present study.

Despite the growing interest in this field and thorough assessments and analyses of international research trends, no studies on the scientific production of sustainability transitions in the construction sector were found. Synthesizing past research findings is an important task for advancing a particular line of research. In this sense, some studies in both the research communities of ST and CI were observed. ST studies addressed sectors such as energy and transport [18] or were more general [14,19], green building studies [20–23], and in sustainability in construction in a broader manner [6,24]. None of the studies addressed these two topics together. Therefore, what is the current state of scientific production on sustainability transition in the construction industry?

To fill this research gap, this study sought to analyze the scientific production on sustainability transition in the construction industry. Specifically, it intended to: (1) verify the publication trend and the main journals in the area; (2) identify the most influential countries, institutions, authors, and works; (3) examine the main relationships of co-authorship, co-citation, and bibliographic coupling; and (4) verify trends in research on the subject. To do so, a bibliometric study was conducted, comprising a sample of 121 articles on the topic extracted from the Scopus and Web of Science databases.

This type of study could help delineate the field and lead to a critical reflection on the research about sustainability transitions, both in empirical and conceptual terms [19].

How the transition research field is dynamic and interdisciplinary and the ongoing efforts to map and analyze the scientific production are important. This enables a reflection on what is published on the topic globally, stimulates discussion about knowledge in the construction area, and allows the evaluation of its evolution, trends, and structuring as a research field. Thus, this type of analysis seeks to facilitate dialogue among academic communities and increase interest in sustainability transitions in communities that have not yet addressed these topics or where the topic is not yet well disseminated—as is the case for the construction research community.

## 2. Literature Review

In this section, we present the fundaments of sociotechnical transitions to sustainability and then exemplify their use in the construction sector.

### 2.1. Sociotechnical Transitions to Sustainability

"Sustainability transitions are long-term, multi-dimensional, and fundamental transformation processes through which established socio-technical systems shift to more sustainable modes of production and consumption" ([19]: p. 956). According to these authors, this transition involves far-reaching changes along different dimensions, including technological, material, organizational, institutional, political, economic, and socio-cultural. In this way, the sociotechnical transitions perspective allows understanding of this process of changes at different levels and domains that interact and align [25], recognizing that companies and technologies are embedded in broader social and economic systems [26].

Then, institutionalized sociotechnical structures, whose fundamental long-term changes lead to their transformation, can ultimately be defined as processes of institutional change with a particular orientation towards technologies [27].

The shift of established sociotechnical systems involves various actors who can adopt a proactive or incumbent position, such as the government, the scientific community, actors from the financial system, the supply network, social groups, and users [28], either supporting or opposing the transition [28,29]. The interaction among these actors occurs based on alignment (cooperation) or confrontation (competition), depending on whether the innovations reinforce or confront the established interests, accepted patterns, and shared beliefs. In response, social groups mobilize to pressure public sector agents into establishing laws and regulations that favor their interests, either hindering—or even preventing—the emergence of innovations or promoting their adoption and diffusion.

By understanding this transition as a co-evolutionary process between artifacts (technologies), people (agents), and institutions (or rules), transition studies help to understand the reasons why some cleaner technologies are not spreading rapidly [30]. This is particularly important for new technologies that are fundamentally different from established technological structures and needs the development of supportive structures that legitimize and stabilize the emerging technology [31].

According to [19], in the ST research field four strands of investigation stand out: *Multi-Level Perspective* (MLP); *Strategic Niche Management* (SNM); *Transition Management* (TM); Technological Innovation Systems (TIS). They can be seen as models to interpret the transition or policy tool in order to govern it, each covering particular aspects of the whole process and complementing each other. Despite their complementarity, each one of them was developed separately and can be applied individually. In the following, we briefly describe the four strands of investigation used by transitions researchers.

Strategic Niche Management (SNM) was first introduced in the late 1990s in the Netherlands by Arie Rip, initially as a research model and later as a policy tool for managing technological innovations [32]. Building on technology and innovation studies and the history of technology and social construction of technology, SNM suggests that sustainable innovation journeys can be facilitated by creating technological niches [33,34]. Niches can be defined as protected spaces for certain applications of a new technology [34]. These protected spaces allow experimentation with the co-evolution of technology, user practices, and regulatory structures [33]. They function as "incubation rooms" for radical novelties and provide spaces for learning processes, for example those about technical specifications, user preferences, public policies, and symbolic meanings, among others [35]. For [33,34], SNM focuses on the role of internal niche processes, such as learning, networks, vision, and the relationship between local projects and global sets of rules that guide the behavior of actors. However, empirical findings have shown that the analysis of these internal niche dimensions needs to be complemented with attention to external processes [33], which can be achieved using a MLP.

The MLP originates from a group of Dutch researchers from the University of Twente and encompasses institutional, sociological, legal, and technical variables [32]. The multi-level perspective is grounded in the works of Rip, Schot, and Kemp [26,36] but gained more popularity with Geels. According to [37], the theoretical roots of MLP are the social construction of technology, evolutionary economics, and neoinstitutional theory. The MLP explains transitions through the interaction of three different levels: niches, regimes, and landscapes [38]. This accommodates a multi-level analysis where niche represents a micro-level, regime a meso-level, and landscape a macro-level.

The concept of a niche is incorporated from SMN. Regime refers to the set of social functions and widely accepted rules by different actors or groups and was built on the concept of "technological regimes" by Nelson and Winter, according to [38]. In other words, the most institutionalized way of performing a social function [39]. This set of rules provides guidance and coordination for the activities of different groups that interact and promote regime stability [37]. The landscape can be defined as an external structure

or context comprising macroeconomic, macropolitical, and cultural factors that shape activities, such as oil prices, economic growth, major crises (e.g., wars, emigration), and issues related to environmental preservation [38]. These factors are beyond the direct influence of actors and cannot be changed at will [35]. In this way, landscapes do not determine actions but provide deep structural "force gradients" that make some actions easier than others [37]. In other words, whereas the landscape can reinforce the incumbent regime by contributing to its stability and permanence, it can also be a source of pressure for its transformation, creating windows of opportunity for niche innovations to emerge [39].

According to [37], in the MLP, interactions between these levels can give rise to emerging ideas, artifacts, and innovations that can be incorporated or even become dominant in a particular context, sometimes systematically modifying the regime and other times merely reconfiguring it. Thus, the MLP aims to understand the nature, characteristics, and operating models of sociotechnical regimes, the sources of stability, and the conditions under which systems change, particularly the processes through which transitions to different sociotechnological systems occur [40]. In other words, it allows us to understand the processes of technological transition and systemic innovation and their contribution to sustainability.

Transition Management (TM) originated in the Netherlands through the work of Rotmans and Kemp in the 2000s, according to [32]. It was initially applied to understand and explain the impact of governance processes on transition [25] and later operationalized as a model to guide policy practice [32]. TM is theoretically rooted in policy, political sciences, sociology, and complexity sciences [41]. Based on complexity theory, TM proposes a new form of governance in a multi-level model that, unlike previous models that take technology as a starting point, primarily focuses on social systems [42]. It is structured into four levels: strategic—which seeks to structure the problem, forecast, and establish the transition arena; tactical—which seeks to develop coalitions, visions, and transition agendas; operational—which seeks to mobilize actors and execute projects and experiments; and reflexive—which seeks to evaluate, monitor, and learn [43]. Analogue to the notion of niche in SNM, this model takes the concept of the transition arena as the starting point of analysis. "The transition arena as a new institution for interaction can be considered a meta instrument for transition management and facilitates interaction, knowledge exchange and learning between the actors" [41]. TM emphasizes the policy mix, governance arrangements, and the government's effectiveness in fostering transitions.

Finally, the Technological Innovation Systems (TIS) approach was developed in Sweden as part of a research program led by Bo Carlsson and Stankiewicz in the early 1990s, according to [32]. With theoretical roots in systems of innovation and technology, according to [44], "A technological system is defined as a dynamic network of agents interacting in a specific economic/industrial area under a particular institutional infrastructure and involved in the generation, diffusion, and utilization of technology". The initial focus of TIS was to understand the contribution of technological innovation to the economic growth of countries; however, more recent research has started to consider new technologies as the key cores of sociotechnical transitions [19]. To ST studies, a technological innovation system can be seen as an application context, in which radical innovations emerge and mature. This is similar to the niche conception (in SNM and MLP), but broader, and might encompass niches [45]. In other words, TIS can be seen as actors and organizations forming arrangements for technological innovation cooperation. This arrangement functions under a particular institutional infrastructure as the essential driver behind the generation, diffusion, and utilization of technological innovation [45].

Table 1 summarizes the main characteristics of SNM, MLP, TIS, and TM.

After presenting the fundaments of ST research, in the following we intend to conceptualize the construction sector through these lenses.

**Table 1.** Main characteristics of SNM, MLP, TIS, and TM.

| SNM | MLP | TIS | TM |
|---|---|---|---|
| **Theoretical Background** | | | |
| Technology and innovation studies, history of technology and social construction of technology. | Social construction of technology, evolutionary economics, and neoinstitutional theory. | Systems of innovation and technology. | Policy, political sciences, sociology, and complexity sciences. |
| **Transitions dynamics** | | | |
| Technological innovation must be nurtured in sociotechnical niches to then spread and reach the regime; the diffusion occurs through institutional dynamics that favor interaction. | Interaction of three different levels: niches, regimes, and landscape. | The overall performance of the innovation system is a function of the degree of alignment of policies and expectations among the different actors in the cooperation arrangement. | Free-market incentives do not lead to sustainability; thus, transitions need governance. Transition to sustainability is a centrally coordinated process, with the government as a key actor in promoting the transition through policy mix and appropriate intervention. |
| **Emphasis of analysis** | | | |
| Interactions between niche (micro) and regime (meso) for the creation of new technologies. | The role of actors (agency) and institutional dynamics (drivers and barriers) within and between different sociotechnical levels (niche, regime, and landscape) in the processes and trajectories of sustainability transition. | Actors and organizations forming arrangements for technological innovation cooperation. | Policy mix, governance arrangements, and the government's effectiveness in fostering transitions. |
| **Main drivers of transition** | | | |
| Niche formation, sociotechnical alignment, the development of expectationsand social desirability, and the availability of complementary resources. | Include the SNM drivers and regime instability, regime selection, niche breakthrough, and niche full development. | Knowledge development and diffusion; market formation; entrepreneurial experimentation; influence on the direction of search; resource mobilization; legitimation and development of positive externalities. | Government enforcement and proper policy mix towards transitions. |

*2.2. The Construction Sector through the Lenses of Sociotechnical Transitions to Sustainability*

The construction sector is part of a widespread economic activity that relies on and stimulates numerous production and service activities [46]. It involves a dispersed chain of participants, such as owners, contractors, architects, engineers, suppliers, regulatory bodies, financing, and administration, each following different business processes and pursuing distinct and often conflicting objectives [47]. The construction sector is responsible for the built-up environment (infrastructure and buildings), including its conception, design, construction, operation and maintenance, and demolition/renovation [48].

The process of transforming the natural environment into the built-up environment entails various impacts throughout the life cycle of a project [49]. Buildings, in this sense, materially represent the entropic processes of society and technology on the environment and the effects of transitions (social and technological) to be overcome, here expressed in the condition of environmental degradation and social inequalities [50].

However, the main studies on sustainability construction do not cover these major dilemmas. The literature emphasizes the construction stage in mitigating the impact of intervention through resource rationalization [51,52]. Some studies propose sustainability indicators and reveal low or unbalanced performance among sustainable dimensions [53–55]. Others address environmental certifications [56,57] or apply life cycle assessment to account

for different consumptions such as energy [53] or other impacts in economic, social, and environmental dimensions [6].

The review articles that cover the research trends in green buildings also highlight the technical side. They emphasize the efficiency of technologies and their financial benefits [21], sustainability performance analysis, design assessment, material and products, rating systems and certifications, optimization, and advanced technologies [23]. Moreover, they highlight green building energy technologies, including building structures, materials, and energy systems [22], alternative materials, sustainable construction management, and recycling and waste reduction [20]. Topics such as codes, regulations, and policies [23], as well as social sustainability in construction management [20], are less frequent.

Against this background, it is evident that these studies lack a comprehensive approach that addresses institutional, sociological, legal, and technical factors [32] and is capable of inducing a transition from conventional construction to green and sustainable construction by integrating the product, design, user, and organizational, social, and environmental dimensions [9]. This is needed to understand the adoption of new materials, methods, processes, and innovative technologies that will transform the sector [58] and lead to sustainability, which is emphasized in ST studies.

Therefore, to elucidate the use of ST concepts in the CI, we take the study of [3] as an example. This study examined the "coevolution through interaction" of Innovative Building Technologies (IBTs) though a case study of modular integrated construction and robotics in Hong Kong. The authors identified that previous studies on IBTs were mostly concerned with elaborating on the technology itself regarding the technical specifications or managerial requirements, whereas an understanding of how IBTs evolve together was absent. Then, the study used the MLP to capture the broader picture of how niche innovations emerge and cause changes in the construction industry. To delve further into the detail of the interactions among niche innovations, the study used a typology model of interaction that categorized them as competition, symbiosis, or neutralism.

The importance of that study lies in the identification of the interactions between technologies, enabling a better understanding of the transition pathway to shape the transformation of the construction regime. The authors identify the three types of interaction modes conceptualized (competition, symbiosis, and neutralism) and describe their co-evolution.

For example, modular integrated construction and automated/robotic on-site factories significantly differ in construction solutions. Whereas the first minimizes on-site activities (off-site construction), the second focuses on automating more construction activities on-site. At the same time, both target industrializing the traditional fragmented construction processes and alleviating the demand for on-site labor. Therefore, an increased share for one of them in the market could form a direct threat to the other, i.e., a bounded competition interaction.

The use of robotic excavators/exoskeletons/drones in modular integrated construction could evolve in an adaptive and neutral manner. They can boost productivity by changing the operational methods in construction, whereas their target markets and corresponding resources do not significantly overlap. Finally, the authors identity three scenarios of reinforced symbiosis interaction between the technologies. First, modular integrated construction could unilaterally benefit from applying robots during off-site manufacturing. Robotics could enhance the productivity, cost efficiency, quality, and safety of the module production performance in factories. Second, the robots are integrated into the modular integrated construction site for module installation based on cyber-physical systems. Third, robots can be embedded into the building lifecycle stages from design to demolition, and modular integrated construction leverages robotics networking to realize full automation and information synchronization.

In sum, the use of the sociotechnical transition approach in the study provides complementary insights into how niche innovations in the construction industry emerge and evolve from the aspects of application scale, diffusion speed, and potential to enable

systems changes. This complements the studies that focus on technical specifications or managerial requirements. The authors use the concept of niche innovations to encapsulate the innovative building technologies (modular integrated construction, automated/robotic, robotic excavators/exoskeletons/drones), the core idea of SNM, and an important level of analysis to MLP used in that study. The authors describe the potential transition dynamics, another important variable of transitions studies that was described in Section 2.1. Finally, based on the potential drivers of transition conceptualized in SNM and MLP, the authors propose implications for policy and organization strategies.

Based on these theoretical foundations, a comprehensive review of the studies on sustainability transitions in the construction sector was conducted following the research method detailed in the next section.

## 3. Method

To conduct our bibliometric review, data were mined from Scopus and Web of Science on 5 January 2023, specifically targeting journals and articles published up to 2022. Scopus and Web of Science were selected due to their comprehensive coverage of peer-reviewed papers from reputable publishers and the accessibility of bibliometric data for conducting in-depth analyses. Figure 1 illustrates the steps of the research conducted, which is detailed in the following sections.

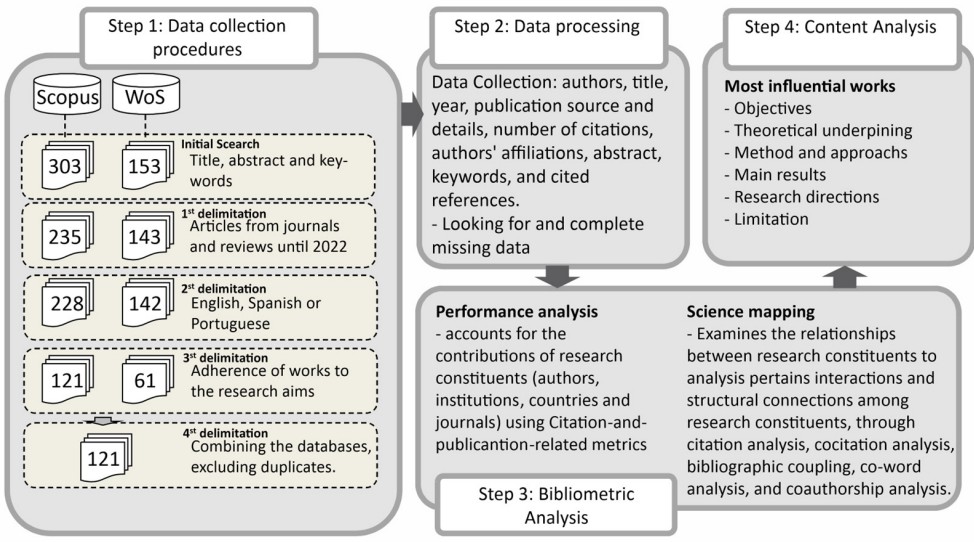

**Figure 1.** Research steps.

The first step was the data collection from Scopus and Web of Science. The query string used in this study was TITLEABS-KEY (("Sustainability transitions" OR "sustainable transitions" OR "sustainable sociotechnical transitions" OR "Socio-Technical Perspective" OR "sociotechnical regime" OR "Socio-Technical Regime" OR "Socio-technical transitions" OR "Transition pathway" OR "Transition theory" OR "technological change" OR "multilevel perspective" OR "MLP" OR "Strategic niche management" OR "SNM" OR "Transition management" OR "TM" "Technological Innovation Systems" OR "TIS") AND ("construction industry" OR "Construction Sector" OR "building sector" OR "building construction" OR "housing" OR "Built environment")), which yielded 303 documents from the Scopus database and 153 from the Web of Science database. A first delimitation was applied to filter only papers and reviews published in journals, which yielded 235 documents in Scopus and 142 in Web of Science. A second delimitation was that the publication language was restricted to English, Spanish, and Portuguese, which yielded 228 documents in Scopus and 142 in Web of Science. The filter was applied due to the proficiency of the present authors for the content analysis of documents in the last step of the research. The third filter applied was the adherence of works to the research aims. This was conducted through a thorough

screening process by reading the title, abstract, and full text and yielded 121 documents in Scopus and 61 in Web of Science. For example, the paper "Testing the central prediction of housing tenure transition models" was excluded, despite the terms "housing" and "transition" being contained within the title. The paper "proposes a tenure choice model, based on tenure transition theory principles" was beyond the scope of the research on transition in the construction sector. Finally, we removed the documents duplicated between/within the two databases, leaving a collection of 121 articles for analysis.

In the second step, tabulation and data processing were carried out. After importing the two databases, inconsistencies and missing data were fitted. Duplicated information or spelling errors were solved to increase the consistency of the data and analysis. We scrutinized 636 keywords to rename certain keywords with synonyms, such as relabeling "Multilevel perspective", "multi-level perspective", and "multi-level perspective (MLP)" to "multi-level perspective (MLP)". This process reduced the number of keywords from 636 to 408. A similar process was carried out to correct any discrepancies in the names of countries and institutions. We also looked for and completed missing data. For example, the lack of keywords in an article from the collection was completed by accessing the entire document. These procedures enabled us to have a consistent bibliometric analysis, as detailed in the following.

In the third stage, a bibliometric analysis was carried out, consisting of two sub-stages according to [59]: (1) performance analysis and (2) scientific mapping. For performance analysis, a summary of publications (authors, institutions, countries, and journals) was carried out using publication and citation metrics to describe the quantitative evolution over time. This led to following results: (1) annual publication trends, (2) most productive journals, (3) most prolific authors, (4) leading countries and institutions, and (5) identification of main works. These results are presented in Sections 4.1–4.5.

In turn, in our scientific mapping step, network analyses based on co-authorship connections, co-citations, keyword co-occurrences, and bibliographic coupling [60] were executed with the support of VOSviewer© software (version 1.6.14) to investigate the complex network of global research collaboration and shed light on recent trends in the scholarly work [60]. VOSviewer is an open-source software tool. We selected it due to the fact that it offers sufficient features for visualizing bibliometric networks and for scientifically mapping the literature. Finally, in the construction management research domain, VOSviewer has been widely used to map knowledge [23].

For co-authorship networks, we set the minimum number of documents by the author to 3 and the minimum number of citations to 0, which yielded 14 authors in the network. Some of them were not connected; for this reason, they were not shown in Section 4.6. A further step was also performed: analyzing the themes formed by these co-authorship networks through a content analysis of the papers, as presented in Section 4.6. Similarly, for co-citation networks of publications, we set the minimum number of citations by documents to 3, which yielded 49 items in network, as presented in Section 4.7. For bibliographic coupling, we set the minimum number of shared references in documents to 20, which yielded 42 items in the network. Finally, for author keyword co-occurrence, we set the minimum number of co-occurrences of keywords in documents to 3, which yielded 37 items in the network. Both the bibliographic coupling and author keyword co-occurrence networks are presented in Section 4.8 to explore research trends and directions.

Finally, in the fourth and last stage, a content analysis was carried out on the main works identified through objectives, conceptual references (the use of frameworks such as MLP, SNM, TM, or TIS), methodology, results, limitations, and future research. These results are presented in Section 4.5, which is a deep analysis of the main works.

## 4. Results and Discussions

The final sample consisted of 121 works from 2001 to 2022, 100% of which were present in the Scopus database. A total of 50.4% of the articles were at the intersection of the Web of Science and Scopus, and none were exclusive to the Web of Science. Figure 2

shows the composition of articles by database. The works in the sample were written by 276 authors (97 as first author and 179 as co-author). These authors are affiliated with 122 institutions located in 21 countries. The papers were published in 64 journals with a total of 3610 citations.

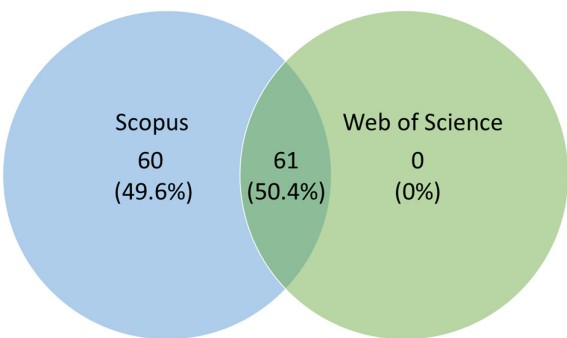

**Figure 2.** Article sampling by database.

### 4.1. Annual Publication Trends

Figure 3 presents the annual publication trends. The line represents the cumulative sum of publications over the years and the bars indicate the quantity of publications per year. An average per quadrennium was also presented to portray the growth rate over the years.

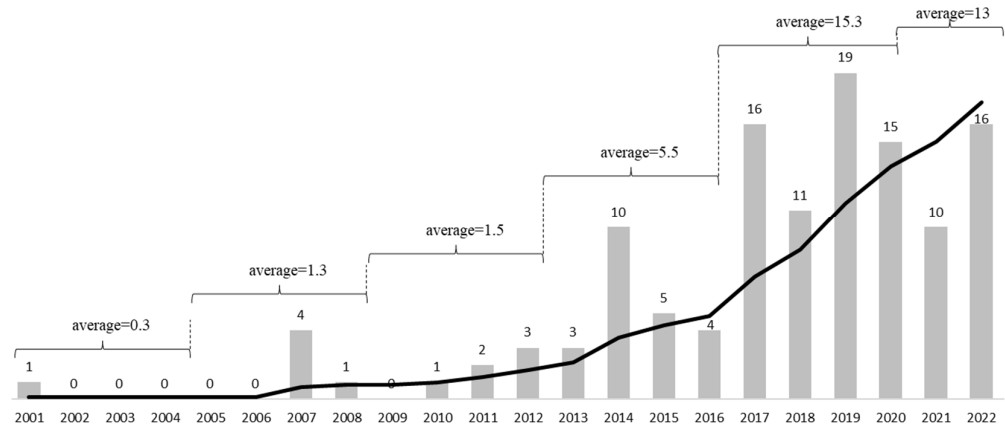

**Figure 3.** Annual Publication Trends.

The annual publication trends captured by the sample show an inconsistency in the number of publications between 2001 and 2009. From the year 2010, there is frequency and volume in the publications, with a growing trend in the number of publications, from an average of 1.5 for the four-year period from 2009 to 2012 to an average of 13 for the period 2020–2022.

The first registered article is authored by [9], the fourth most cited in the sample, which confers upon them the character of a seminal author. The article was published in the journal *Technology Analysis and Strategic Management*, which is ranked among the top 10 most productive journals in the sample. In the second four-year period, five works were published. These studies have accumulated 931 citations, of which, three of the studies, ref. [61–63], account for 70% of the citations in this period. In the period from 2009 to 2012, six articles were published with a total of 426 citations; three of the studies, ref. [64–66], accounted for 78% of the citations. In the period from 2013 to 2016, 22 articles were published, 5 of which accounted for 41% of the citations: [67–71]. For the period 2017 to 2020, the period with the most publications (61), six works account for 38% of the citations: [72–77]. Finally, in the last two years, 26 articles were published with a total of 138 citations, including the works of [78–82].

The remarkable rise in publications in recent years reinforces the importance of research on the transition to sustainability in the construction industry. This is in line with the reinforcement by the European Union, which highlights the sector as one of the three key sectors to address the challenges of climate change [12], positioning it at the top of the intergovernmental agenda [83] and on the radar of the IPCC (Intergovernmental Panel on Climate Change) and EEA (European Environment Agency) [16].

In addition, publications were observed in the main journals in the field of transitions (such as *Environmental Innovation and Societal Transitions*), as well as in the fields of innovation, production, energy, regional studies, geography, and Architecture, Engineering, and Construction (AEC). In this sense, the effort made in the present research proves promising, as it will organize the main information about this bibliographic production.

*4.2. Most Productive Journals*

Table 2 presents the most productive journals and the number of papers published over time.

**Table 2.** Most productive journals and the number of papers published over time.

| Rank | Journal | 2001 to 2004 | 2005 to 2008 | 2009 to 2012 | 2013 to 2016 | 2017 to 2020 | 2021 to 2022 | Total | % of Papers |
|---|---|---|---|---|---|---|---|---|---|
| 1 | *Journal of Cleaner Production* | 0 | 0 | 0 | 3 | 6 | 3 | 12 | 9.9 |
| 2 | *Environmental Innovation and Societal Transitions* | 0 | 0 | 0 | 1 | 9 | 0 | 10 | 8.3 |
| 3 | *Building Research and Information* | 0 | 1 | 1 | 2 | 1 | 2 | 7 | 5.8 |
| 4 | *Sustainability (Switzerland)* | 0 | 0 | 0 | 1 | 3 | 3 | 7 | 5.8 |
| 5 | *Construction Management and Economics* | 0 | 0 | 0 | 1 | 3 | 1 | 5 | 4.1 |
| 6 | *Technological Forecasting and Social Change* | 0 | 1 | 0 | 2 | 1 | 0 | 4 | 3.3 |
| 7 | *Energy Policy* | 0 | 1 | 0 | 1 | 2 | 0 | 4 | 3.3 |
| 8 | *Global Transitions* | 0 | 0 | 0 | 0 | 2 | 1 | 3 | 2.5 |
| 9 | *Technology Analysis and Strategic Management* | 1 | 1 | 0 | 0 | 1 | 0 | 3 | 2.5 |
| | Zone 1 (Σ 9 journals with least 3 papers) | 1 | 4 | 1 | 11 | 28 | 10 | 55 | 45.4 |
| | Zone 2 (Σ 11 journals with 2 papers) | 0 | 0 | 2 | 3 | 11 | 6 | 22 | 18.2 |
| | Zone 3 (Σ 44 journals with 1 paper) | 0 | 1 | 3 | 8 | 22 | 10 | 44 | 36.4 |
| | Total (64 journals) | 1 | 5 | 6 | 22 | 61 | 26 | 121 | 100.0 |

Initially, a significant concentration of documents is observed in the first nine journals, whereas the remaining areas exhibit high dispersion, i.e., out of the total number of journals (64), 45.5% of the papers are published in nine journals (14.1% of the number of journals). These journals are: *Journal of Cleaner Production* (ISSN 0959-6526, Impact Factor (IF) 11.072), *Environmental Innovation and Societal Transitions* (ISSN 2210-4224, IF 9.377), *Building Research and Information* (ISSN 0961-3218, IF 5.322), *Sustainability (Switzerland)* (ISSN 2071-1050, IF 3.889), *Construction Management and Economics* (ISSN 0144-6193, IF 4.048), *Technological Forecasting and Social Change* (ISSN 1873-5509, IF 10.884), *Energy Policy* (ISSN 0301-4215, IF 7.576), *Global Transitions* (ISSN 2589-7918, IF 0.300), and *Technology Analysis and Strategic Management* (ISSN 1465-3990, IF 4.250). The latter journal published the first paper on transitions in the field of construction in the sample.

Out of the total, it can be observed that 15 journals (23%) are related to Architecture, Engineering, and Construction (AECO). However, only two of those journals (*Building Research and Information* and *Construction Management and Economics*) are among the top nine most productive journals, with 29 publications (24% of the total). This indicates the low engagement of research in sustainability transitions in the AECO field.

*4.3. Most Prolific Authors*

Table 3 presents the key authors, citations, and institutions based on the number of articles published as first author, co-author, and total. The table shows only authors with two or more papers in the sample.

**Table 3.** Most prolific authors in ST in CI research publications.

| N° | Author | First Author | Co-Author | Total | Total Citations | Current Affiliations |
|----|--------|--------------|-----------|-------|-----------------|----------------------|
| 1 | Kivimaa P. | 5 | 1 | 6 | 163 | Finnish Environment Institute, Finland |
| 2 | Jain M. | 3 | 1 | 4 | 41 | University of Twente, Netherlands |
| 3 | Moore T. | 3 | 1 | 4 | 19 | RMIT University, Australia |
| 4 | Killip G. | 3 | 0 | 3 | 31 | University of Oxford, UK |
| 5 | Gibbs D. | 2 | 2 | 4 | 79 | University of Hull, UK |
| 6 | Horne R. | 2 | 2 | 4 | 26 | RMIT University, Australia |
| 7 | O'Neill K. | 2 | 2 | 4 | 176 | Cardiff University, UK |
| 8 | Chang R. D. | 2 | 0 | 2 | 45 | University of Adelaide, Australia |
| 9 | Edmondson D.L. | 2 | 0 | 2 | 131 | University of Sussex, UK |
| 10 | Enker R. A. | 2 | 0 | 2 | 12 | Curtin University, Malaysia |
| 11 | Fastenrath S. | 2 | 0 | 2 | 15 | University of Cologne, Germany |
| 12 | Hagbert P. | 2 | 0 | 2 | 10 | KTH—Royal Institute of Technology, UK |
| 13 | Hemström K. | 2 | 0 | 2 | 32 | Linnaeus University, Sweden |
| 14 | Jiang H. | 2 | 0 | 2 | 14 | University of Sheffield, UK |

In general, these fourteen authors (or 5.1% of the 276 authors) have published 34 papers (or 28% of the total 121 papers in analysis) and together have accumulated 679 citations (19.3% of the total 3610 citations). On the other hand, 262 authors (94.9%) contributed to the 87 analyzed articles. This indicates that only a small "elite" of authors has published more than one article in the field.

Additionally, it was found that the seminal or most cited authors are from the top three countries that produce the most research works on the topic. These results are corroborated with leading countries presented in the following section.

### 4.4. Leading Countries and Institutions

Table 4 highlights the 18 countries where articles were produced.

**Table 4.** Production by country (articles by authorship).

| N° | Country | First Author | Co-Author | N° | Country | First Author | Co-Author |
|----|---------|--------------|-----------|----|---------|--------------|-----------|
| 1 | UK | 36 | 44 | 10 | Norway | 3 | 0 |
| 2 | The Netherlands | 18 | 39 | 11 | Malaysia | 2 | 32 |
| 3 | Australia | 17 | 48 | 12 | Italy | 1 | 3 |
| 4 | China | 11 | 28 | 13 | Poland | 1 | 2 |
| 5 | Finland | 9 | 25 | 14 | Canada | 1 | 1 |
| 6 | Germany | 6 | 8 | 15 | France | 1 | 1 |
| 7 | USA | 5 | 9 | 16 | Switzerland | 1 | 0 |
| 8 | Denmark | 5 | 6 | 17 | Thailand | 0 | 2 |
| 9 | Sweden | 4 | 8 | 18 | New Zealand | 0 | 1 |

The UK stands out as the most productive country, with 36 articles. Combined with The Netherlands and Australia, these three countries account for 71 works, representing 58% of the total number of publications. Figure 4 shows the map of countries with publications in the sample and the number of papers published as the first author.

The map of countries shows that the sample consists of articles from four out of the five inhabited continents (America, Europe, Asia, and Oceania). No studies from institutions in Africa and Latin America were identified. The emphasis is on European countries, as expected, and several variables pertaining to Europe can explain these results, such as: (1) high urbanization rate and population density, which leads to a transition to sustainability at the city and regional level, with buildings and infrastructure playing a crucial role; (2) well-defined weather seasons, which require buildings to have heating technologies in winter and cooling technologies in summer, consuming a significant amount of energy and emitting more greenhouse gases than desired for zero-carbon or zero-energy housing;

(3) greater environmental awareness among the population, making the market for green buildings and infrastructure more promising; (4) higher productivity in science, technology, and innovation, fostering interaction between academia and industry to generate crucial innovations for the transition; and (5) greater institutional maturity, with political support from various backgrounds, such as the European Parliament and multilateral organizations such as the United Nations (UN).

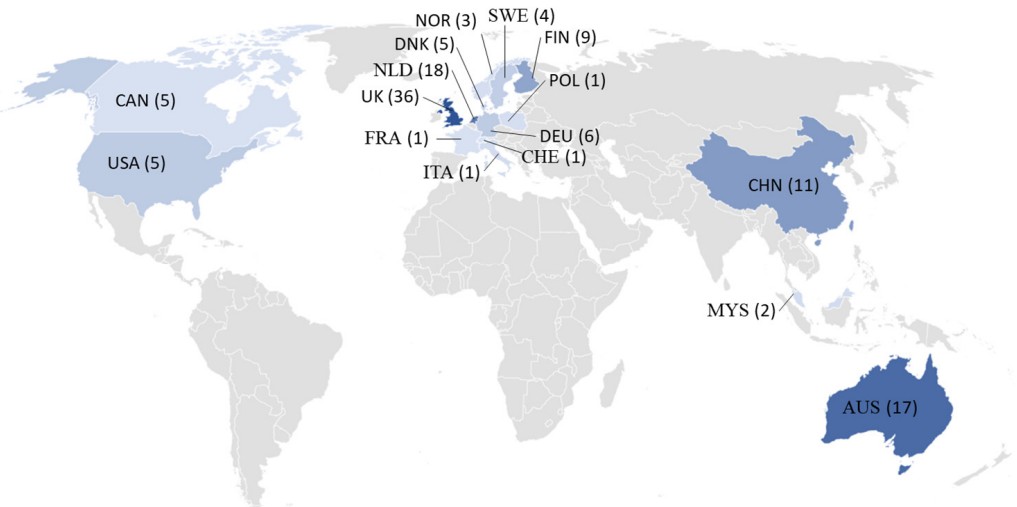

**Figure 4.** Map of countries with first author.

The absence of scientific production from the Global South (or developing countries in general), including Latin America and Africa, suggests that in these regions the debate around sustainability transition in the construction sector is not yet sufficiently developed, except in China. This implies the absence of transitions in the sector that can be published in peer-reviewed journals in the Web of Science and Scopus databases. The discussions about transitions in the Global South are ongoing, with much evidence from other sectors in these countries, such as the smart mobility transition in the city of Curitiba, Brazil [84] and MLP analysis of EasyTaxi in Colombia [85], both from Latin America. From Africa, ref. [86] explores energy transitions in South Africa and [87] explores urban sustainability transitions, comparing the cities of Curitiba and Accra, the latter is the capital and largest city of Ghana. The studies are examples from among others studies from these regions. More insights for research and policy on ST in developing countries can be accessed in [88].

China's performance can be explained by the rapid process of industrialization and the corresponding urbanization of regional hubs, as well as the governance model, in which the government plays a crucial role in directing transition efforts. According to [89], to promote green housing development, the Chinese government has issued various policies and regulations into the Chinese housing market. The Chinese government's ambition is to promote energy efficiency in the new urban building sector by requiring 50% of new urban buildings to be green buildings by 2020.

Finally, it is worth noting that the language filter (English, Portuguese, and Spanish) can also bias the sample, despite the minimal reduction achieved through this filter (235 to 228 in Scopus and 143 to 142 in Web of Science).

A total of 122 academic institutions were identified, out of which 80 appear as the affiliation of first author in at least one paper. Table 5 presents the 13 academic institutions that have three or more articles with their affiliated authors listed as the first author.

It can be observed that 6 out of the 13 most productive institutions are in the UK, followed by the Netherlands with three institutions, and Finland and Australia with one each. In this list, only RMIT University is not located in Europe, confirming the predominance of Europe in the sample. In sum, there is a gap in transition studies concerning various regions, especially the Global South.

**Table 5.** The most productive academic institutions.

| Nº | Institution | First Author | Country |
|----|-------------|--------------|---------|
| 1 | University of Sussex | 9 | UK |
| 2 | RMIT University | 7 | Australia |
| 3 | Utrecht University | 6 | The Netherlands |
| 4 | University of Twente | 5 | The Netherlands |
| 5 | University of Hull | 4 | UK |
| 6 | Finnish Environment Institute | 4 | Finland |
| 7 | Delft University of Technology | 4 | The Netherlands |
| 8 | University of Helsinki | 4 | Finland |
| 9 | KTH—Royal Institute of Technology | 4 | UK |
| 10 | Aalto University | 3 | Finland |
| 11 | University of Oxford | 3 | UK |
| 12 | University of Leeds | 3 | UK |
| 13 | University of Sheffield | 3 | UK |

*4.5. Main Works*

Table 6 presents the works with the highest impact based on the number of citations.

**Table 6.** Main works in ST in the CI research publications.

| Nº | Author | Title | Journal | Total Citations | % of Citations | % Cumulated of Citation |
|----|--------|-------|---------|-----------------|----------------|-------------------------|
| 1 | Smith (2007) [61] | Translating sustainabilities between green niches and socio-technical regimes | *Technology Analysis and Strategic Management* | 549 | 15.2 | 15.2 |
| 2 | Brown e Vergragt (2008) [62] | Bounded socio-technical experiments as agents of systemic change: The case of a zero-energy residential building | *Technological Forecasting and Social Change* | 180 | 5.0 | 20.2 |
| 3 | Berkhout et al. (2010) [64] | Sustainability experiments in Asia: Innovations shaping alternative development pathways? | *Environmental Science and Policy* | 176 | 4.9 | 25.1 |
| 4 | Rohracher (2001) [9] | Managing the technological transition to sustainable construction of buildings: A socio-technical perspective | *Technology Analysis and Strategic Management* | 169 | 4.7 | 29.8 |
| 5 | Edmondson, Kern e Rogge (2019) [72] | The co-evolution of policy mixes and socio-technical systems: Towards a conceptual framework of policy mix feedback in sustainability transitions | *Research Policy* | 131 | 3.6 | 33.4 |
| 6 | Kivimaa et al. (2017) [73] | Experiments in climate governance—A systematic review of research on energy and built environment transitions | *Journal of Cleaner Production* | 106 | 2.9 | 36.3 |
| 7 | Beerepoot and Beerepoot (2007) [63] | Government regulation as an impetus for innovation: Evidence from energy performance regulation in the Dutch residential building sector | *Energy Policy* | 105 | 2.9 | 39.2 |
| 8 | Noailly (2012) [65] | Improving the energy efficiency of buildings: The impact of environmental policy on technological innovation | *Energy Economics* | 91 | 2.5 | 41.7 |
| 9 | Kivimaa et al. (2019) [74] | Passing the baton: How intermediaries advance sustainability transitions in different phases | *Environmental Innovation and Societal Transitions* | 78 | 2.2 | 43.9 |
| 10 | Gibbs e O'Neill (2015) [68] | Building a green economy? Sustainability transitions in the UK building sector | *Geoforum* | 71 | 2.0 | 45.9 |
| | Zone 1 (10 papers with more than 70 citations) | | 9 journals | 1656 | 45.9 | 45.9 |
| | Zone 2 (25 papers with 30 to 70 citations) | | 19 journals | 1073 | 29.7 | 75.6 |
| | Zone 3–1 (76 papers with 1 to 30 citations) | | 46 journals | 881 | 24.4 | 100 |
| | Zone 3–2 (10 papers without citations) | | 8 journals | 0 | 0 | 100 |
| | Total | | | 3610 | 100 | 100 |

Initially, we can observe a high concentration of citations in the 10 main works, whereas the remaining zones exhibit high dispersion. Furthermore, the works with the highest impact were also published in the main journals described in Section 4.2 and authored by the key authors mentioned in Section 4.3.

The analysis of the works with highest impact allowed us to understand the relevant aspects of the scientific production on sustainability transitions in the construction sector. Then, we performed a content analysis of the top 10 papers as detailed in the following.

Firstly, it is noticeable that most of the research adopts a qualitative approach, with most studies employing case studies (four papers), literature reviews (three papers), or theoretical essays (one paper). Only two studies were quantitative. An interesting aspect of the qualitative studies was the proposition of frameworks, as seen in [62,64,72–74]. In these studies, the construction sector was used to illustrate the theoretical propositions presented, either through case studies conducted by the authors [62,72] or included in literature reviews [62,73,74].

Regarding the quantitative studies conducted by [63,65], both focused on analyzing the impact of environmental policies on technological innovations aimed at improving building energy efficiency.

On the other hand, the theoretical essay in [9] sought to problematize the construction sector from a sociotechnical perspective. It aimed to clarify the differences that sustainability-oriented innovation and technology studies can bring to a reflection on sustainable construction.

Considering the four research strands (MLP, SNM, TM, and TIS) stand out among the ten works with the highest impacts, only three studies partially or fully employed MLP in combination with SNM literature [64,68] and one used a combination of MLP, SNM, and TM [73]. This reinforces the complemental character among the research strands. For example, ref. [68] explores the green building sector in the UK as an innovation niche and its interaction with the building and construction industry as sociotechnical regimes, the micro- and meso-levels of the MLP, respectively. To understand the niche development in more depth, the authors used the SNM. They explored how the UK government's policy efforts to encourage green building have led to niche activities challenging the existing building regime.

The absence of studies applying TIS suggests more difficulty in forming networks of actors and organizations for technological innovation cooperation. This aligns with the conclusion in [90], which argues that the sector operates as a loosely coupled system, favoring productivity in projects while hindering innovation.

Regarding the references used for transitions, there is homogeneity around the seminal authors in the field, such as Geels, who popularized the multi-level perspective, and the authors who underpin them [26,36]. Seminal authors for the SNM [33,34] and TM [91] strands were also present. Finally, the suggestions for future work provided by the authors reveal thematic areas, including the interaction between sociotechnical levels, policies for transition in the sector, climate governance and experiments, and intermediaries.

In sum, the analysis of the main works suggest that the field is still in its early stages and requires more research for a comprehensive understanding, i.e., the conceptual limits and boundaries have yet to be developed.

### 4.6. Collabortion Network

Figure 5 presents the co-authorship networks. Out of the 276 authors, 14 have published three or more works. Four authors are not connected to any other and are not displayed in the presented network, which is composed of 10 authors. The colors represent the working groups, and the size of the circle reflects the number of documents authored by each researcher.

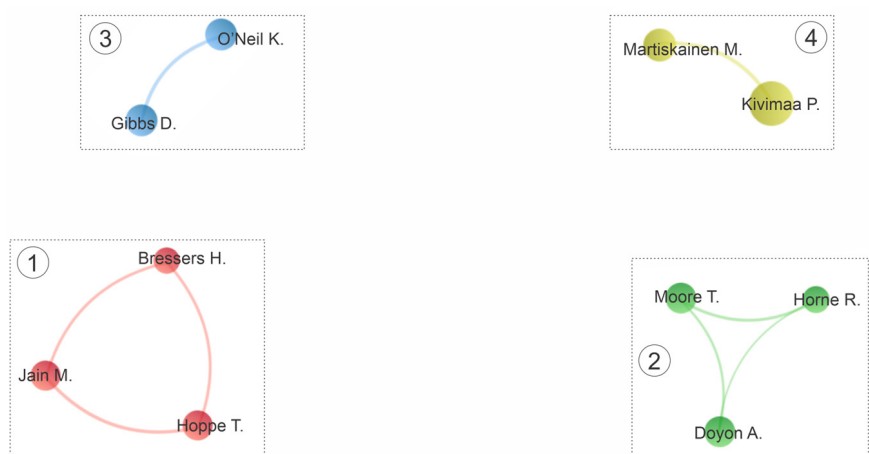

**Figure 5.** Co-authorship network.

Four collaboration clusters were identified; these are isolated from each other, revealing the ongoing fragmentation within the publication network in the field of transitions to sustainability in the construction sector. Essentially, these clusters have formed around authors who have co-published more than one work together, as detailed in Table 7.

**Table 7.** Co-authorship network and themes.

| Cluster | Co-Authorship Network | Themes |
|---|---|---|
| 1 (red) | Jain, Hoppe, and Bressers from the University of Twente, Netherlands. | Energy and green buildings: net-zero energy buildings in India [92,93] and low energy green building in Singapore and Delhi [94]. |
| 2 (green) | Trivess Moore, Andréanne Doyon, and Ralph Horne from the RMIT University VIC, Australia. | Sustainable housing: sustainable housing innovation in Australia [95–98] and study of policy development in Australia and comparisons with the EU, UK, USA, and California to zero emission housing [99]. |
| 3 (blue) | David Gibbs and Kirstie O'Neill from the University of Hull, UK. | Green building: sociotechnical transitions in the green building sector in the UK [67,68,100,101]. |
| 4 (yellow) | Paula Kivimaa from the University of Sussex, UK. | Other: typology of experiments in climate governance [73]; zero carbon homes in the UK [76]; intermediaries in sustainability transitions [74]; Finnish policies on building energy efficiency transition [102]. |

According to the findings of the co-authorship network and themes, sustainability transitions in the construction industry can be viewed as a disruptive phenomenon and have mainly taken place in developed countries. Cluster 1 reveals a trend in developing countries, such as India, indicating an earlier stage of transitions in the construction industry, with the observation of niche formation. In the works by Jain and collaborators [92–94], they explore the formation of the net-zero energy buildings niche in India. This result corroborates the low production observed in the sample from countries of the Global South that was discussed in Section 4.4.

*4.7. Theoretical Basis*

Out of the 7615 references cited in the sample, 49 of them were cited at least three times; these are represented in Figure 6. The co-citation network highlights the authors cited in the publications, revealing the most explored theoretical basis by the authors in the sample [59].

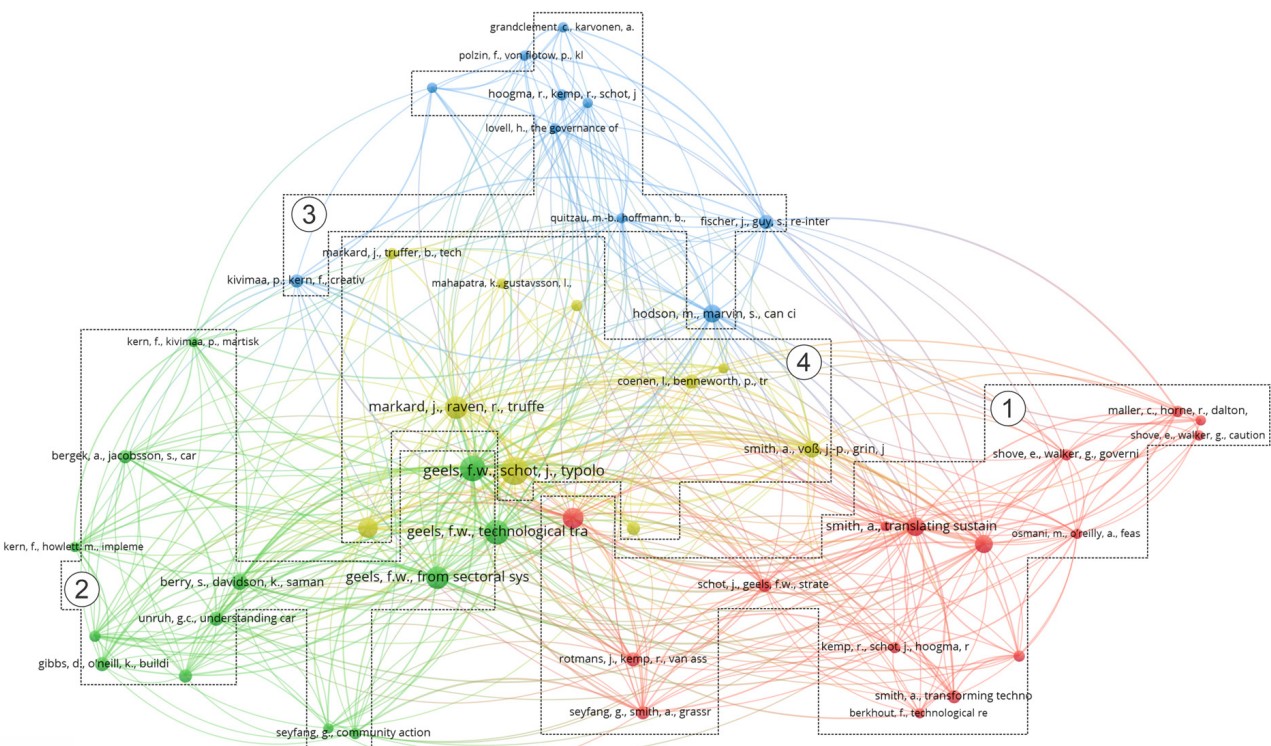

**Figure 6.** Co-citation networks of publications (at least three papers). The numbers from 1 to 4 denote the formation of four clusters.

One can observe the existence of four clusters that group around the following lines: (1) fundaments of transitions to sustainability; (2) dynamics of transition to sustainability; (3) governance of transition to sustainability; and (4) epistemology of sustainability transitions research.

Cluster 1, highlighted in red and titled "fundaments of transitions to sustainability", includes 16 papers with 83 citations. Within this cluster, ref. [30,33,61,103–106] coalesce 63% of the cluster's citations. Reference [30] argues that changes result from the articulation of selection pressures acting on the regime and the provision and coordination of resources for its adaptation. Reference [61] noted how the translation of regime problems has an important constitutive effect on the creation of niches. Reference [103] discusses seven social science ontologies (rational choice, evolutionary theory, structuralism, interpretivism, functionalism, conflict and power struggle, and relationism). Reference [104] brings Transition Management (TM) to the discussion and proposes a new form of governance in a multi-level model. Reference [105] proposes a new research and policy agenda for sustainable development based on grassroots innovations. Finally, ref. [106] discusses some of the challenges faced by a theory of regime transformation and how historical research on alternative technology can contribute to this.

Cluster 2, "dynamics of transition to sustainability", shown in green, has 13 papers with 96 citations. Noteworthy works include [35,38,107], which together were cited 50 times (52% of the cluster's citations). Reference [38] presents a multi-level perspective of sociotechnical transitions that explains transitions through the interaction of three different levels: niches, regimes, and landscape. This is one of Geels' most popular works; Geels is an important theorist for the MLP and ST in a broader sense. Reference [35] bases the dynamic interactions between systems, actors, and the rules of regimes to explain stability and change in a multi-level perspective. Reference [107] dissects the concept of a niche and proposes three functional properties in relation to broader transition processes: shielding, nurturing, and empowerment.

Cluster 3, "governance and transition to sustainability", in blue, has 10 papers that were cited 42 times by the sample articles. The central works include [91,108–110], which together were cited 24 times (57% of the cluster's citations). Reference [108] reflects on the capacity of cities to shape sociotechnical transitions. Reference [109] analyzes the influence of regulations in promoting low-carbon buildings. Reference [110] develops an analytical framework, extending the approach of the functions of the technological innovation system. Reference [91] discusses strategic niche management for sustainable transport. Underlying these studies is the concern regarding how governments can promote the introduction of new technologies and niche development, which is at the heart of SNM and directly addressed in [91,108].

Cluster 4, "epistemology of sustainability transitions research", shown in yellow, includes 10 papers with 81 citations among the sample articles. The central works are [19,37,39,111], which together were cited 56 times (69% of the cluster's citations). Reference [37] proposes one of the most recognized typologies for sociotechnical transition pathways. References [19,39,111] offer critical reflections on the development of sustainability transitions research. These reflections help to make terminologies, objectives, and methods more coherent within the research field [112]. In this regard, ref. [39] elaborates on the challenges of operationalizing the multi-level perspective of sociotechnical transitions (MLP). Reference [111] presents responses to seven criticisms of the MLP: (1) lack of agency, (2) operationalization of regimes, (3) bias towards bottom-up change models, (4) epistemology and explanatory style, (5) methodology, (6) landscape as a residual category, and (7) flat ontologies versus hierarchical levels. Finally, ref. [19] offers a view on the emerging research field and the transition to sustainability and it prospects. The authors identify the intellectual contours of this field, conducting a review of the basic conceptual frames.

In sum, these results provide researchers with a foundation for key concepts and challenges in structuring research on transitions from epistemological, methodological, and conceptual perspectives. It is a useful starting point to study ST. It reveals those who sought to understand regime changes and the importance of experiments and niche development for research in the construction sector. Underlying these studies, the predominance of MLP usage is observed; however, studies on niches and experimentation expand niche dynamics through SNM. Furthermore, mobilizing actors and organizations for forming arrangements for technological innovation cooperation, as proposed in the TIS model, still appears to be a challenge in the construction sector. This highlights the role of intermediaries, as already portrayed in the literature. Another challenge seems to be advancing studies that emphasize social systems, as proposed in the TM, and even the role of governments in governance for sectoral transition.

### 4.8. Research Trends and Directions

Bibliometric analysis can provide valuable insights into research trends and directions in ST in the CI. For doing so, we use bibliographic coupling and author keyword co-occurrence networks. By analyzing bibliographic coupling, we can understand the periodic or current development of themes in the research field [59]. The bibliographic coupling relationship between two works is greater the more references they share, indicating the theoretical and/or methodological proximity between works [113]. To explore the existing or future relationships between topics in a research field, we used a keyword co-occurrence network [59].

Figure 7 presents the bibliographic coupling network of the sample. To improve visualization, it was chosen to present articles that had 20 or more shared references. This represents approximately 30% of shared articles based on the average number of references in the sample, which is 63 citations (7615 references per 115 papers). This criterion led to a network with 42 papers that had 2129 citations (59% of the total citations of the sample articles).

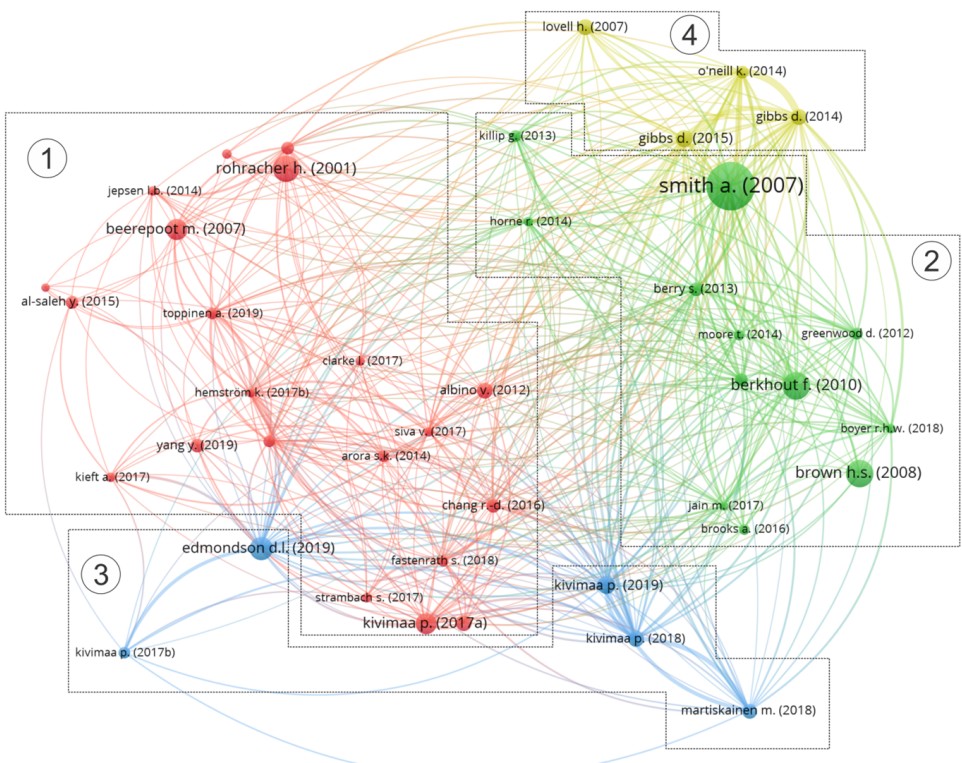

**Figure 7.** Bibliographic coupling (at least 20 references). The numbers from 1 to 4 denote the formation of four clusters.

In this way, four clusters were identified and are detailed as follows: (1) changes in the sociotechnical regime of construction industry; (2) experiments in transition to sustainability; (3) intermediaries and public policies for transition; and (4) development of green niches.

Cluster 1, "Changes in the sociotechnical regime of construction industry", in red, groups 21 works that have a total 1178 citations (33% of the total citations in the entire sample). Noteworthy works include [66,69,77,114,115], which together have a total of 552 citations (15% of the total citations in the sample). Except for [114], the cluster's works address the civil construction regime. Reference [69] analyzes the transition to sustainability in the Chinese construction industry, based on the practices and behaviors of leading builders. Reference [114] investigates the performance of everyday domestic practices using building monitoring. Reference [77] analyzes the barriers that inhibit the transition to sustainability in the Australian construction industry. Reference [115] analyzes the architects' perceptions about innovation in the Swedish construction industry. Finally, ref. [66] investigates how intercompany relationships are changing as the Italian construction sector moves towards green buildings.

Cluster 2, "Experiments in the transition to sustainability", highlighted in green, groups 12 works with 636 citations (18% of the total citations in the entire sample). References [61,64,70,116] accumulate 402 citations (11% of the total citations in the sample). Reference [70] presents a practical demonstration of the government's vision for sustainable living by exploring how structural change at the regime level can arise from the incubation of ideas and experiences at the niche level under the influence of landscape pressures. Reference [64] addresses the theme of experiments in the transition to sustainability, concluding that: (i) the actor networks in sustainability experiments are heterogeneous in their composition; (ii) the regimes and landscapes are relatively fluid, rather than stable; (iii) the diffusion of learning and technology between experiments, niches, and regimes faces multiple institutional barriers; and (iv) there is a limited understanding of how global knowledge links influence the development and growth of sustainability experiments.

Reference [116] explores instances of buildings with clean energy solutions. Reference [61] focuses on niche–regime interaction, highlighting the importance of the concept of "sociotechnical translations" and tries to understand how the translation processes operate over time. The author points out how the translation of the regime's problems has an important constitutive effect on the creation of niches.

Cluster 3, "Intermediaries and public policies for transition", in blue, groups five works, ref. [72,74–76,102], that together have a total of 200 citations (6% of the total citations in the entire sample). Reference [75] conducts a systematic review of the case studies involving low-energy buildings in Europe, focusing on intermediaries. Reference [76] conducts empirical research focusing on zero-carbon homes in the UK. Additionally, on the theme of intermediaries, ref. [74] proposes an integration of existing conceptual models on transition dynamics and phases and a typology of transition intermediaries to examine how intermediaries advance in transitions at different phases. Reference [102] analyzes the potential to facilitate a zero-carbon transition from a mixture of Finnish public policies, based on a customer-oriented evaluation from a frontier actor perspective. Lastly, ref. [72] explores how policy mixes influence sociotechnical change and how changes in the sociotechnical system also shape the evolution of policies. The authors propose a new conceptual framework to conceptualize the coevolutionary dynamics of policy mixes and sociotechnical systems. The authors demonstrate the framework's applicability using the example of the policy mix for zero-carbon homes in the UK.

Finally, Cluster 4, "Developments of green niches", groups four works with 115 citations (3% of the total citations in the entire sample) distributed among [67,68,100,117]. Reference [67] explores the development of green entrepreneurship and its potential role in the change towards a green economy in the construction sector of the UK. They concluded that although the green economy and the green construction sector present coherent identities with consensual and consistent practices, these are not unanimous when business models vary and that there are significant contradictions within so-called green construction practices. The same authors, in 2015, reflected on the capacity of the green discourse to promote radical changes for the transition to sustainability in the construction sector [68]. Reference [117], as already mentioned in the co-citation analysis, addresses the difficulties of strategic niche management in practice, using the development of low-energy housing in the UK to illustrate some limitations of this technology change model. Finally, ref. [100] analyzes the sociotechnical transitions in the green construction sector, focusing on the role of green entrepreneurs in effecting change.

According to the findings the bibliographic coupling, in Cluster 1, it can be observed that the studies address the transition from the perspective of different actors (user practices, architects, and professionals from construction companies) to analyze the types of barriers perceived as relevant and how they contribute to resistance to change and path dependency in the building construction sector. In a complementary manner, Cluster 2 includes articles that emphasize the importance of experiments to promote transition in the sector. Such experiments help reveal costs and increase understanding of the economic risks associated with innovations and the compliance to a norm of adhering to that which is considered to be proven, barriers to innovation, and consequently to the transition in the construction sector, as highlighted by [115]. Cluster 3 reveals the importance of intermediary institutions, as well as public policies, in the transition process within the sector. Finally, Cluster 4 grouped works that emphasized the development of green niches. Across these clusters, there is an emphasis on low-energy buildings.

Now, we explore the existing or future relationships between topics in the research field using the keyword co-occurrence network [59]. Figure 8 presents the author keyword co-occurrence network. Out of a total of 408 keywords, 37 were used at least three times, as shown in Figure 8. These keywords are associated in four clusters that are detailed in the sequence.

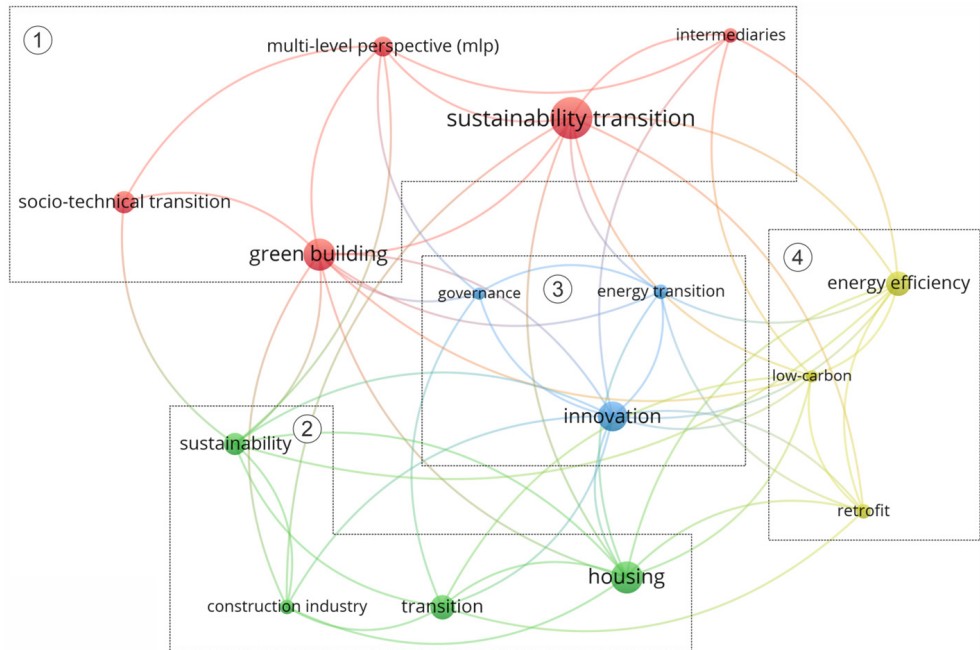

**Figure 8.** Bibliometric map of author keyword co-occurrence (at least three occurrences).

In Cluster 1, highlighted in red, there is a connection between the themes of green buildings and the perspective of sociotechnical transitions; the cluster is formed by the words green building, multi-level perspective (MLP), sociotechnical transition, sustainability transition, and intermediaries. Cluster 2, highlighted in green, consists of the keywords construction industry, housing, sustainability, and transition and can be designated as sustainability in housing construction. Cluster 3, highlighted in blue, comprises the keywords governance, energy transition, and innovation. It can be designated as governance to energy transition. Finally, Cluster 4 brings together the keywords energy, low carbon, and retrofit, which can be designated as strategies for energy transition, as retrofit implies the readjustment of buildings and constitutes a strategy to achieve an energy transition in the building stock. This cluster includes works that examine the energy performance of innovations implemented in buildings, with a focus on technical variables.

The keyword co-occurrence network results show the existing relationships between topics that are again around low-energy buildings and green buildings. Moreover, the findings of the bibliographic coupling and author keyword co-occurrence networks are coherent and aligned.

From the above discussion, we can outline some research trends:

- The green building concept is more disseminated in the studies from the construction industry and presents a coherent identity to guide discussions and strategies for transition in the sector. Furthermore, it appears that the so-called green construction practices are already more developed and advanced than what is observed in other communities of practice;
- It seems productive to identify and map the niches and experiments in the construction sector that lead to transition. The lessons from these experiences should be addressed in a way that enhances the structures that promote learning and increases the dissemination of new technologies in the sector. For this purpose, joint efforts by the government–industry–universities are essential to promote the technological transition;
- The research explores the role of intermediary organizations as a means for unlocking transitions in the sector. The literature for the TIS provides insights into this research avenue, such as in the works [31,118];

- Review the policy and regulations and the role of government to guide the transition in the sector. The literature of TM offers insights into this research avenue, bringing in the concept of the transition arena and taking social systems as a starting point.

## 5. Conclusions

The present study aimed to analyze the scientific literature on the transition to sustainability in the construction industry. To this end, a bibliometric study was carried out on the Scopus and Web of Science databases. This review identified the evolution of scientific production and the top journals, institutions, nations, and authors contributing to this field and highlighted a significant increase in publications since 2017. Furthermore, through network analysis, the collaboration among authors, the theoretical bases of the field, and the main research themes and trends were revealed. This mapping is the main contribution of this work and provides an overview of the scientific production on the transition to sustainability in the construction industry.

The scientific production on the transition to sustainability in the construction sector is diverse but relatively recent, indicating that the field is still in its early stages and requires more research for a comprehensive understanding. Moreover, the low dissemination of the theme in the architecture, engineering, and construction community implies that the research findings may not have reached a wider audience within the industry, potentially limiting its real-world impact.

The huge number of publications attesting to ST in the CI can be viewed as a disruptive phenomenon that has mainly taken place in developed countries. It is naïve to consider that actors in different regions have equal capabilities when it comes to performing the tasks required to transition to sustainability. Therefore, an interesting topic for future research could be identifying the niche in formation and the conditions needed to drive the transition in the Global South.

A significant portion of the accessed literature has a theoretical profile and uses the construction sector for illustrative purposes, indicating a potential need for more empirical research and practical case studies. This fact enables the opportunity to problematize the construction sector and its contribution to research on transitions to sustainability, as well as to clarify the differences that studies of innovation and technology oriented towards sustainability can bring to a reflection on sustainable construction, as identified in [9]. Thus, what unique characteristics does the sector have that allows for a deeper understanding of aspects of the transition? One of them is the power of incumbent actors who have had dominant technologies since the 20th century (reinforced concrete, for example) [119]. Therefore, future research could explore in more depth how and why these powerful incumbent actors participate (or not) in a sustainable transition.

Another aspect is that the construction sector's embeddedness in an institutional environment with weak institutions poses challenges to unlocking the transition to sustainability, as conflicting interests among supporting actors can hinder progress. Therefore, understanding how to "unlock" the transition in the construction sector can significantly benefit transition studies.

Despite the findings, the study has limitations that might serve as the foundation for future studies. First of all, even though the bibliometric method identifies the framework of a knowledge corpus, it cannot substitute additional review approaches, such as meta-analysis and qualitative literature review. Therefore, a more detailed categorization process, based on the content of the works, could be performed. This effort would allow the debate of the results to be expanded, identifying, for example, the main barriers to transition in the sector, which is suggested as future work. Thus, the goal of this study is to serve as the first step toward a more complete evaluation of the literature on ST in the CI. Secondly, since knowledge on ST in the CI is still in its early research stages, the construct's conceptual limits and boundaries have yet to be developed. Thirdly, the search was limited to journal articles and excluded other types of publications such as conference proceedings, which may contain valuable information and be more up to date. Nonetheless, this decision

was made to ensure the quality and reliability of the sources used in this review. Journals provide comprehensive and in-depth coverage of specific research topics and undergo rigorous peer-review processes to ensure the quality and credibility of published articles.

Therefore, while acknowledging the limitations of this bibliometric review, we believe that the presented analysis provides a comprehensive overview that can and will inspire future researchers to build upon our work and explore the additional future research suggested, thus uncovering even more valuable insights into this important topic.

**Author Contributions:** Conceptualization, L.F.C., J.C.L., A.O.d.F.e.S. and J.d.P.B.N.; methodology, L.F.C. and A.O.d.F.e.S.; software, L.F.C.; validation, J.C.L., A.O.d.F.e.S. and J.d.P.B.N.; formal analysis, L.F.C. and A.O.d.F.e.S.; investigation, L.F.C.; resources, L.F.C., J.C.L., A.O.d.F.e.S. and J.d.P.B.N.; data curation, L.F.C. and A.O.d.F.e.S.; writing—original draft preparation, L.F.C., J.C.L., A.O.d.F.e.S. and J.d.P.B.N.; writing—review and editing, L.F.C., J.C.L., A.O.d.F.e.S. and J.d.P.B.N.; visualization, L.F.C., J.C.L., A.O.d.F.e.S. and J.d.P.B.N.; supervision, J.C.L. and J.d.P.B.N. All authors have read and agreed to the published version of the manuscript.

**Funding:** This research and this publication is funded by Fundação ASTEF (Fundação de Apoio a Serviços Técnicos e Fomento a Pesquisa), as well by FUNCAP, CAPES, CNPQ, UFC and UESPI.

**Data Availability Statement:** No new data was created or analyzed in this study. Data sharing is not applicable to this article.

**Conflicts of Interest:** The authors declare no conflict of interest.

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
