# Peer review of "Sustainability Transitions in the Construction Sector: A Bibliometric Review"

_sustainability, doi:10.3390/su151712814_

Round 1

Reviewer 1 Report

1.In line 101, the format of the reference is not consistent with others.

2.The sentence in line 157 has mistakes in grammar.

3.Figure 2 has mistakes, as the statistics 0 and 0% are not correct.

4.In Tables 2-4, the word main author is ambiguous. Does it refer to the first author, corresponding author, or others? The word Co-autor also has a spelling error.

5.In Table 3, Hong Kong cannot be listed as a country, but should be a region.

6.In Figure 4, the meaning of the numbers was not explained, and some of the top ranked countries in Table 3 did not indicate the numbers. Taiwan is a part of China, and the color of Taiwan island should be consistent with that of Chinese Mainland.

7.In the last row of Table 5, the first three cumulative numbers (64, 3610, 100) are not equal to the sum of the four numbers in their respective upper four rows.

8.In Table 6, please translate the part of Theme into English.

9.Figures 6 and 7 are repeated.

10.The font in Figures 6-8 is too small.

11.It is lack of research contribution. Further, this article lacks detailed explanations for future research, which reduces the research value.

Moderate editing of English language required.

Author Response

Author's Response to Reviewers' Comments

Dear Editor and Reviewer

We would like to genuinely thank the reviewer for the constructive and meaningful feedback. All recommendations were considered in this last version of the manuscript. We, the authors, remain open for any further clarification.

Regarding questions from Reviewer 1

[Authors] We reviewed the suggestions 1 to 10 and fix them in the text. The author's response to reviewer comment 11 is addressed separately, as highlighted below.

1.In line 101, the format of the reference is not consistent with others.

2.The sentence in line 157 has mistakes in grammar.

3.Figure 2 has mistakes, as the statistics 0 and 0% are not correct. >>> não entendi.

4.In Tables 2-4, the word “main author” is ambiguous. Does it refer to the first author, corresponding author, or others? The word “Co-autor” also has a spelling error.

5.In Table 3, Hong Kong cannot be listed as a country, but should be a region.

6.In Figure 4, the meaning of the numbers was not explained, and some of the top ranked countries in Table 3 did not indicate the numbers. Taiwan is a part of China, and the color of Taiwan island should be consistent with that of Chinese Mainland.

7.In the last row of Table 5, the first three cumulative numbers (64, 3610, 100) are not equal to the sum of the four numbers in their respective upper four rows.

8.In Table 6, please translate the part of Theme into English.

9.Figures 6 and 7 are repeated.

10.The font in Figures 6-8 is too small.

11. It is a lack of research contribution. Further, this article lacks detailed explanations for future research, which reduces the research value.

[Authors] We reviewed the highlights of research contributions to clarify this point. We would like to highlight that study: (1) provides insightful information about the current state of TS in CI; (2) revealed that the field is still in its early stages and requires more research for a comprehensive understanding; (3) identify the low dissemination of the theme in the architecture, engineering, and construction community which implies that the research findings may not have reached a wider audience within the industry, potentially limiting its real-world impact; (4) reveals that transitions in construction industry have mainly taken place in developed countries and provide some reflections about this; (5) indicating a potential need for more empirical research and practical case studies due of the theoretical profile of literature accessed that uses the construction sector for illustrative purposes; (6) we also outline some research trends; and (7) proposes some future research.

Reviewer 2 Report

Interesting paper, however in the conclusions it is suggested to address the limitations, such as the absence of a more detailed categorization process, with more detailed design of further steps

Editing of the English language should be applied

Author Response

Author's Response to Reviewers' Comments

Dear Editor and Reviewer

We would like to genuinely thank the reviewer for the constructive and meaningful feedback. All recommendations were considered in this last version of the manuscript. We, the authors, remain open to any further clarification.

Regarding questions from Reviewer 2

Interesting paper, however in the conclusions it is suggested to address the limitations, such as the absence of a more detailed categorization process, with a more detailed design of further Steps

[Authors] We have improved the discussion around the paper's limitations and enriched its content. First of all, we highlight the limitation in the nature of the used methodology. The science mapping's methodical and quantitative approach helps identify the framework of a knowledge corpus, but it cannot substitute additional review approaches, such as meta-analysis and qualitative literature review. Therefore, a more detailed categorization process, based on the content of the works, can be performed.

This effort would allow the debate of the results to be expanded, identifying, for example, the main barriers to transition in the sector, which is suggested as future work.

Thus, the goal of this study is to serve as the first step toward a more comprehensive evaluation of the literature on ST in CI. Secondly, since knowledge of ST in CI is still in its early research stages, the concept's conceptual limits and boundaries have yet to be fully developed. Thirdly, the search was limited to journal articles and excluded other types of publications such as conference proceedings, which may contain valuable and up-to-date information. Nonetheless, this decision was made to ensure the quality and reliability of the sources used in this review. Journals provide comprehensive and in-depth coverage of specific research topics and undergo rigorous peer-review processes, ensuring the quality and credibility of published articles.

Reviewer 3 Report

1. Abstract

1.1. Lack of specific details about the findings or key aspects of the bibliometric review, making the abstract too general.

1.2. Use of vague language, which could be more concise and specific to enhance clarity.

1.3. Incomplete mention of the methodology used for the bibliometric review, leaving out important information about the study's approach.

1.4. Limited discussion of the significance or potential implications of the findings, missing the opportunity to highlight the practical importance of the study.

1.5. Ambiguous language that could be clearer and more directly related to the study's outcomes.

2. Introduction

2.1. Lack of specific citations to support the claims about the Construction Industry being traditional, low-tech, and producing high rates of waste and environmental pollution.

2.2. Repetitive information, particularly regarding the strategic importance of the construction sector for sustainability transition.

2.3. Ambiguous language in describing the current state of the literature on sustainability transitions in the construction sector.

2.4. Lack of clarity on the specific research gap or research question that the study aims to address.

2.5. Inconsistent structure, with the introduction lacking a clear and well-defined outline of the study's objectives and contributions.

3. Literature review

3.1. Lack of specific citations for some claims about the construction sector's impacts and its lack of comprehensive approach in addressing sustainability.

3.2. Repetitive information and lack of clarity in connecting the four strands of investigation (Multi-level Perspective, Strategic Niche Management, Transition Management, and Technological Innovation Systems) to the context of sustainability transitions in the construction sector.

3.3. The need for clearer and more concise explanations of the theoretical foundations of each approach (MLP, SNM, TM, and TIS) to provide a more comprehensive understanding for the reader.

4. Methods

4.1. Lack of clarity and coherence in explaining the steps of the bibliometric review and data analysis, making it difficult for readers to follow the research methodology.

4.2. Missing details or explanations about certain aspects of the bibliometric analysis, such as the specific metrics used to describe the quantitative evolution of publications over time.

4.3. Limited explanation of the relevance and significance of certain analysis methods, such as the use of Lotka's Law, Bradford's Law, and Zipf's Law, and how they contribute to the study.

4.4. Absence of information on the criteria used to select the articles and the specific parameters for data extraction and tabulation.

5. Results and discussions

5.1. The section lacks a critical analysis and interpretation of the results, limiting the depth of understanding and implications of the findings.

5.2. Insufficient details about the methodology used for data collection and analysis may affect the study's transparency and reproducibility.

5.3. Relying solely on citation metrics may not fully capture the significance of research works, overlooking other essential factors.

5.4. The section lacks a clear synthesis of the main findings, making it challenging to identify the key takeaways from the results.

5.5. The section mentions an emphasis on European countries, but it does not discuss potential geographical biases in the sample or variations in other regions.

5.6. Failing to compare findings with existing research in the field limits the context and significance of the study.

5.7. The section briefly mentions collaboration networks without providing in-depth insights into their implications for research dynamics.

5.8. The section does not address study limitations or propose future research directions, which could enhance the overall contribution of the research.

6. Conclusions

6.1. The absence of a more detailed categorization process based on the content of the works limits a deeper understanding of specific aspects of the transition to sustainability in the construction industry.

6.2. The low dissemination of the theme in the architecture, engineering, and construction community implies that the research findings may not have reached a wider audience within the industry, potentially limiting its real-world impact.

6.3. The scientific production on the transition to sustainability in the construction sector is diverse but relatively recent, indicating that the field is still in its early stages and requires more research for a comprehensive understanding.

6.4. The construction sector's embeddedness in an institutional environment with weak institutions poses challenges in unlocking the transition to sustainability, as conflicting interests among supporting actors can hinder progress.

6.5.  A significant portion of the accessed literature has a theoretical profile and uses the construction sector for illustrative purposes, indicating a potential need for more empirical research and practical case studies.

The English in the given text is generally clear and understandable. However, there are a few areas where improvements could be made to enhance clarity, as an example applied to Conclusion section:

  1. "This mapping being the main contribution of this work." --> "This mapping is the main contribution of this work."

  2. "The low dissemination of the theme in the architecture, engineering, and construction community was also verified." --> "The limited dissemination of the theme within the architecture, engineering, and construction community was also confirmed."

  3. "As the actors' support for a transition is conditioned to the articulation of the emerging sociotechnical system..." --> "Since the actors' support for a transition depends on the articulation of the emerging sociotechnical system..."

  4. "Thus, how to 'unlock' the transition in the construction sector seems to be a question from which transition studies can benefit." --> "Therefore, understanding how to 'unlock' the transition in the construction sector can significantly benefit transition studies."

  5. "The absence of a more detailed categorization process, based on the content of the works." --> "The lack of a more detailed categorization process based on the content of the works."

Overall, the text conveys its message effectively, but making these small adjustments can improve its readability and precision.

Author Response

Author's Response to Reviewers' Comments

Dear Editor and Reviewer

We would like to genuinely thank the reviewer for the constructive and meaningful feedback. All recommendations were considered in this last version of the manuscript. We, the authors, remain open to any further clarification.

Regarding questions from Reviewer 3

1. Abstract

1.1. Lack of specific details about the findings or key aspects of the bibliometric review, making the abstract too general.

1.2. Use of vague language, which could be more concise and specific to enhance clarity.

1.3. Incomplete mention of the methodology used for the bibliometric review, leaving out important information about the study's approach.

1.4. Limited discussion of the significance or potential implications of the findings, missing the opportunity to highlight the practical importance of the study.

1.5. Ambiguous language that could be clearer and more directly related to the study's outcomes.

[Authors] We revised the abstract according to the suggested points.

2. Introduction

2.1. Lack of specific citations to support the claims about the Construction Industry being traditional, low-tech, and producing high rates of waste and environmental pollution.

[Authors] We revised the introduction according to the suggested points. In the following, we have added comments in some of them to clarify our revision actions.

[Authors] We use a series of citations for the question above:

  • We replaced the expression “traditional” with “conservative behaviour” based on [2] “Hofman, B.; de Vries, G.; van de Kaa, G. Keeping Things as They Are: How Status Quo Biases and Traditions along with a Lack of Information Transparency in the Building Industry Slow Down the Adoption of Innovative Sustainable Technologies. Sustain. 2022, 14, doi:10.3390/su14138188.”
  • We add “its operational way that is labour-intensive at the construction site [3]” based on “Yang, Y.; Pan, M.; Pan, W. ‘Co-Evolution through Interaction’ of Innovative Building Technologies: The Case of Modular Integrated Construction and Robotics. Autom. Constr. 2019, 107, 102932, doi:10.1016/j.autcon.2019.102932.”
  • We replaced the expression “low-tech” with “low-tech intensity [4]” based “Nykamp, H. A Transition to Green Buildings in Norway. Environ. Innov. Soc. Transitions 2017, 24, 83–93, doi:10.1016/j.eist.2016.10.006.”
  • For producing high rates of waste we use [3,4] Osobajo, O.A.; Oke, A.; Omotayo, T.; Obi, L.I. A Systematic Review of Circular Economy Research in the Construction Industry. Smart Sustain. Built Environ. 2020, doi:10.1108/SASBE-04-2020-0034. and “Backes, J.G.; Traverso, M. Application of Life Cycle Sustainability Assessment in the Construction Sector: A Systematic Literature Review. Processes 2021, 9, doi:10.3390/pr9071248.”
  • We add “These main characteristics are manifested in” to link with “high rates of consumption of raw materials, and environmental pollution” based in [6,7]. “6. Backes, J.G.; Traverso, M. Application of Life Cycle Sustainability Assessment in the Construction Sector: A Systematic Literature Review. Processes 2021, 9, doi:10.3390/pr9071248.” and “7. Liu, H.; Lin, B. Ecological Indicators for Green Building Construction. Ecol. Indic. 2016, 67, 68–77, doi:10.1016/j.ecolind.2016.02.024.”. Additionally, we added “Darko, A.; Chan, A.P.C.; Gyamfi, S.; Olanipekun, A.O.; He, B.J.; Yu, Y. Driving Forces for Green Building Technologies Adoption in the Construction Industry: Ghanaian Perspective. Build. Environ. 2017, 125, 206–215, doi:10.1016/j.buildenv.2017.08.053.”

2.2. Repetitive information, particularly regarding the strategic importance of the construction sector for a sustainable transition.

[Authors] We have reviewed the text and hope to have resolved this issue.

2.3. Ambiguous language in describing the current state of the literature on sustainability transitions in the construction sector.

[Authors] We have reviewed the text and hope to have resolved this issue.

2.4. Lack of clarity on the specific research gap or research question that the study aims to address.

[Authors] We have reviewed the text and hope to have resolved this issue. Despite the growing interest in this field, thorough assessments and analyses of international research trends, no studies on the scientific production of sustainability transitions in the construction sector were found. Synthesizing past research findings is an important task for advancing a particular line of research. In this sense, some studies in both research communities of ST and CI were observed. In ST studies, addressing sectors such as energy and transport [17] or in general [13,18], in green building studies [19–22], and in sustainability in construction in a broader manner [5,23]. None of them addressed these two topics together. So, what is the current state of scientific production on Sustainability Transition in the Construction Industry? To fill this research gap, this study sought to analyze the scientific production on sustainability transition in the construction sector. This type of study could help delineate the field and lead to a critical reflection on the research about sustainability transitions, both in empirical and conceptual terms.

2.5. Inconsistent structure, with the introduction lacking a clear and well-defined outline of the study's objectives and contributions.

[Authors] We have reviewed the text and hope to have resolved this issue.

3. Literature review

[Authors] We revised the literature review section according to the suggested points. We divide the section in two. First, we present the fundaments of Sociotechnical Transitions to sustainability. We provide clearer and more concise explanations of the theoretical foundations of each approach (MLP, SNM, TM, and TIS) and summarize them in Table 1. After presenting the fundamentals of ST research, we elucidate the use of ST concepts in CI, through an example from the literature.

3.1. Lack of specific citations for some claims about the construction sector's impacts and its lack of comprehensive approach in addressing sustainability.

3.2. Repetitive information and lack of clarity in connecting the four strands of investigation (Multi-level Perspective, Strategic Niche Management, Transition Management, and Technological Innovation Systems) to the context of sustainability transitions in the construction sector.

3.3. The need for clearer and more concise explanations of the theoretical foundations of each approach (MLP, SNM, TM, and TIS) to provide a more comprehensive understanding for the reader.

4. Methods

[Authors] We revised the method according to the suggested points. In the following, we have added comments in some of them to clarify our revision actions.

4.1. Lack of clarity and coherence in explaining the steps of the bibliometric review and data analysis, making it difficult for readers to follow the research methodology.

[Authors] We improve the description of the bibliometric procedure. We add specific details of our protocol, including the setup of VOSView to network analysis.

4.2. Missing details or explanations about certain aspects of the bibliometric analysis, such as the specific metrics used to describe the quantitative evolution of publications over time.

[Authors] We improve this specific.

4.3. Limited explanation of the relevance and significance of certain analysis methods, such as the use of Lotka's Law, Bradford's Law, and Zipf's Law, and how they contribute to the study.

[Authors] The quoted laws were excluded from the analysis. The contribution was minimal for our study.

4.4. Absence of information on the criteria used to select the articles and the specific parameters for data extraction and tabulation.

[Authors] We improve the description of the bibliometric procedure to fill this gap. The criteria are presented in Figure 1 and detailed in the rest of Section 3. We performed an initial search based on the title, abstract, and keywords. Next, we applied four filters: (1) Articles and reviews from journals until 2022; (2) published in English, Spanish, or Portuguese; (3) adequacy to the research aims; (4) not duplicated in combining the Scopus and Web of Science databases. We also improved the details about the parameters for data extraction and tabulation.

5. Results and discussions

[Authors] We revised the results and discussions according to the suggested points. In the following, we have added comments in some of them to clarify our revision actions.

5.1. The section lacks critical analysis and interpretation of the results, limiting the depth of understanding and implications of the findings.

[Authors] We improve this point. Whenever possible, we articulate the findings with existing research in the area.

5.2. Insufficient details about the methodology used for data collection and analysis may affect the study's transparency and reproducibility.

[Authors] We improve the description of the bibliometric procedure to fill this gap as aforementioned.

5.3. Relying solely on citation metrics may not fully capture the significance of research works, overlooking other essential factors.

[Authors] We agree that there are other factors to capture the significance of research works. Notwithstanding it, we follow the recent literature on bibliometric analysis to set the measures used in our study. According to Donthu et a. (2021), a citation is one of the most important measures of impact and influence, as well as the number of publications and citations per year or per researcher, wherein publication is a proxy for productivity. For those authors, citation analysis is a basic technique for science mapping. In this analysis, the impact of a publication is determined by the number of citations that it receives. The analysis enables the most influential publications in a research field to be ascertained.

Donthu, N.; Kumar, S.; Mukherjee, D.; Pandey, N.; Lim, W.M. How to Conduct a Bibliometric Analysis: An Overview and Guidelines. J. Bus. Res. 2021, 133, 285–296, doi:10.1016/j.jbusres.2021.04.070.

5.4. The section lacks a clear synthesis of the main findings, making it challenging to identify the key takeaways from the results.

[Authors] We add at the end of each section a paragraph summarizing the main findings.

5.5. The section mentions an emphasis on European countries, but it does not discuss potential geographical biases in the sample or variations in other regions.

[Authors] We improve this point as follows. The absence of scientific production from the Global South (or developing countries in general), including Latin America and Africa, suggests that in those regions, the debate around sustainability transition in the construction sector is not yet sufficiently developed. However, the discussion about transitions in the Global South is ongoing, with much evidence from other sectors in these countries, and we provide some examples. Finally, it is worth noting that the language filter (English, Portuguese, and Spanish) can also bias the sample, despite the minimal reduction achieved through this filter (235 to 228 in Scopus, and 143 to 142 in Web of Science).

5.6. Failing to compare findings with existing research in the field limits the context and significance of the study.

[Authors] We improve this point. Whenever possible, we articulate the findings with existing research in the area.

5.7. The section briefly mentions collaboration networks without providing in-depth insights into their implications for research dynamics.

[Authors] We improve this point. The collaboration networks reveal that ST in CI can be viewed as a disruptive phenomenon and has been taking place, particularly in developed countries. Moreover, it reveals a trend in developing countries of an earlier stage of transitions in CI, with the observation of the formation of niches.

5.8. The section does not address study limitations or propose future research directions, which could enhance the overall contribution of the research.

[Authors] We present the limitations of the study and propose future research directions in the concluding section.

6. Conclusions

[Authors] We revised the conclusion section according to the suggested points.

6.1. The absence of a more detailed categorization process based on the content of the works limits a deeper understanding of specific aspects of the transition to sustainability in the construction industry.

6.2. The low dissemination of the theme in the architecture, engineering, and construction community implies that the research findings may not have reached a wider audience within the industry, potentially limiting its real-world impact.

6.3. The scientific production on the transition to sustainability in the construction sector is diverse but relatively recent, indicating that the field is still in its early stages and requires more research for a comprehensive understanding.

6.4. The construction sector's embeddedness in an institutional environment with weak institutions poses challenges in unlocking the transition to sustainability, as conflicting interests among supporting actors can hinder progress.

6.5.  A significant portion of the accessed literature has a theoretical profile and uses the construction sector for illustrative purposes, indicating a potential need for more empirical research and practical case studies.

Comments on the Quality of English Language

[Authors] We thank the reviewer for the excellence of having suggested improvements to improve the quality of the English language in our paper. We reread the text looking for and fixing minor inaccuracies in writing and grammar.

The English in the given text is generally clear and understandable. However, there are a few areas where improvements could be made to enhance clarity, as an example applied to the Conclusion section:

  1. "This mapping being the main contribution of this work." --> "This mapping is the main contribution of this work."
  2. "The low dissemination of the theme in the architecture, engineering, and construction community was also verified." --> "The limited dissemination of the theme within the architecture, engineering, and construction community was also confirmed."
  3. "As the actors' support for a transition is conditioned to the articulation of the emerging sociotechnical system..." --> "Since the actors' support for a transition depends on the articulation of the emerging sociotechnical system..."
  4. "Thus, how to 'unlock' the transition in the construction sector seems to be a question from which transition studies can benefit." --> "Therefore, understanding how to 'unlock' the transition in the construction sector can significantly benefit transition studies."
  5. "The absence of a more detailed categorization process, based on the content of the works." --> "The lack of a more detailed categorization process based on the content of the works."

Overall, the text conveys its message effectively, but making these small adjustments can improve its readability and precision.

Thank you again to the reviewer for the questions, we believe this new version of the paper achieves a really upper level of contribution.

Round 2

Reviewer 1 Report

Thanks the authors to make lots of revisions. However, there are still two points that should be carefully handled.

1. In Figure 4, the color of Taiwan island should be consistent with that of Chinese Mainland. We know that science knows no borders, but scientists do. I urge to revise this Figure, or please just delete Figure 4, as the information of Figure 4 have already been shown in Table 4. For the same reason, in Table 4, the data for England, Scotland, and Wales should be merged into UK data.

2. In the last row of Table 6, why the cumulative number 64 are not equal to the sum of 9, 19, 46 ,8 should be explained, and why the cumulative number 100 are not equal to the sum of 46, 30, 23 should also be explained.

Minor editing of English language required, such as CO2 should be CO2.

Author Response

Author's Response to Reviewers' Comments - Round 2

Dear Editor and Reviewer

We would like to genuinely thank the reviewer for the constructive and meaningful feedback. All recommendations were considered in this last version of the manuscript. We, the authors, remain open for any further clarification.

Regarding questions from Reviewer 1

1. In Figure 4, the color of Taiwan island should be consistent with that of Chinese Mainland. We know that science knows no borders, but scientists do. I urge to revise this Figure, or please just delete Figure 4, as the information of Figure 4 have already been shown in Table 4. For the same reason, in Table 4, the data for England, Scotland, and Wales should be merged into UK data.

[Authors] In order to address this review's recommendations, we have implemented the following updates in the text:

  • In line 441 we replaced “21 countries” with “19 countries” attending to the suggestion by the reviewer to merge England, Wales, Ireland, and Scotland in the UK.
  • In line 442, Table 4, we excluded the countries that were merged in the UK and updated the enumeration from 21 to 18 countries.
  • In line 445 we replaced “68 works, representing 56%” with “71 works, representing 58%” due to the suggestion by the reviewer to merge England, Wales, Ireland, and Scotland in the UK.
  • In line 448, we replaced Figure 4 to rectify the color of the Taiwan island in alignment with the Chinese Mainland

2. In the last row of Table 6, why the cumulative number 64 are not equal to the sum of 9, 19, 46 ,8 should be explained, and why the cumulative number 100 are not equal to the sum of 46, 30, 23 should also be explained."

[Authors] In order to address this review's recommendations, we have implemented the following updates in the text:

  • We have eliminated the summation of the journal count, a factor that had previously posed challenges in terms of interpretation and was a concern effectively addressed in response to the reviewer's inquiry. The total count of journals in the 'Journals' column exceeds 64, which is the overall number of journals within the sample. This discrepancy arises due to our classification of articles into different zones based on their citation counts. As a result, a single journal might be encompassed in multiple zones. For example, the journal 'Technological Forecasting and Social Change' has articles spanning zones 1, 2, and 3. This means that papers from this specific journal encompass citation ranges of more than 70 citations, 30 to 70 citations, and 1 to 30 citations, respectively.
  • In line 500 (Table 6), we have increased the number of decimal places in the columns 'Percentage of Citations' and '% Cumulative Citations' to ensure that the total sum precisely adds up to 100%.

[Authors] Finally, we revise the author contribution section, and we reread the text looking for and fixing minor inaccuracies in writing and grammar.

Reviewer 3 Report

No further comments.

I find it satisfactory.

Author Response

Author's Response to Reviewers' Comments - Round 2

Dear Editor and Reviewer

The Reviewer 3 assigned “No further comments.”

We would like to genuinely thank the reviewer for the constructive and meaningful feedback throughout the evaluation process. We, the authors, remain open for any further clarification.
